# Potential Synergistic Inhibition of *Enterococcus faecalis* by Essential Oils and Antibiotics

Stanley John, Jeung Woon Lee, Purushottam Lamichhane, Thanhphuong Dinh, Todd Nolan and Thomas Yoon *

LECOM School of Dental Medicine, 4800 Lakewood Ranch Blvd., Bradenton, FL 34211, USA;
sjohn97534@dmd.lecom.edu (S.J.); jwlee@lecom.edu (J.W.L.); lamichhanepurru@gmail.com (P.L.);
kdinh@lecom.edu (T.D.); tnolan@lecom.edu (T.N.)
* Correspondence: tyoon@lecom.edu; Tel.: +1-941-405-1509

**Abstract:** Recurrent infections after root canal treatments often involve *Enterococcus faecalis*, a microorganism closely associated with therapy failures due to its biofilm production, survival in nutrient-deprived conditions, and antibiotic tolerance. Essential oils (EOs), which display antimicrobial and antibacterial properties, exhibit inhibitory effects on the growth of many microorganisms including *E. faecalis*. This study assessed the in vitro efficacy of combining 5% antibiotics (kanamycin 2.5 mg/mL, streptomycin 2.5 mg/mL, gentamicin 1.5 mg/mL, and ampicillin 5 mg/mL) with cinnamon (1.25% to 5%) or clove (25% and 50%) EOs in inhibiting the growth of *E. faecalis*, using disk diffusion tests. Disks were treated with EOs-only, antibiotics-only, or EO–antibiotic combinations, placed on BEA agar plates, and incubated for 24 h, and the zones of inhibition were measured. Results showed that EOs (cinnamon and clove) and 5% antibiotics, by themselves, had robust growth inhibition of *E. faecalis* across all tested concentrations. Moreover, combining 5% aminoglycosides (kanamycin 2.5 mg/mL, streptomycin 2.5 mg/mL, and gentamicin 1.5 mg/mL) with 5% cinnamon EO produced significantly enhanced antimicrobial effect than the corresponding 10% antibiotic solution alone. These findings suggest that combining cinnamon EO with aminoglycoside antibiotics can achieve significant inhibition of *E. faecalis* at a lower concentration of antibiotics compared to using a higher dose of antibiotics alone. Further in vivo studies should determine the safety, efficacy, and treatment duration, with the potential to reduce antibiotic dosages and associated toxicity while preventing recurrent infections.

**Keywords:** cinnamon; clove; essential oils; *E. faecalis*; antibiotics; gentamycin; streptomycin; kanamycin; ampicillin; root canal infection

## 1. Introduction

Since their discovery, antibiotics have become essential in successfully combating infectious diseases to improve human health [1,2]. In dentistry, antibiotics are selectively used for orofacial infections of odontogenic and non-odontogenic origins such as dental abscess, pulpal necrosis, periodontal diseases, dental caries, dental trauma, adenoiditis, otomastoiditis, and in prophylaxis [2–5]. Penicillin beta-lactam and amoxicillin, along with clindamycin, are the most widely prescribed oral antibiotic agents in dentistry, prophylactically and for therapeutic use, for their broad-spectrum antimicrobial activity of the former and as an alternate to penicillin allergic responses for the latter [1,6,7]. Along with their health benefits, however, the overprescription of antibiotics is a concern for clinicians and for patients. An analysis of outpatient prescription data from British Columbia showed the dental antibiotic prescription rate increasing by 71.6% between the years 1996 to 2013 [6]. Similar UK and US studies also found that over 65% of antibiotic prescriptions were issued when there was no evidence of spreading infection, and 73% to 85% of antibiotic prescriptions of penicillin beta-lactam and lincosamides for adult and children outpatients were either unnecessary or untimely [8–11].

Microorganisms, such as *actinomyces*, *streptococci*, and *enterococci* (*E. faecalis*), that produce dental biofilms, complex communities of microorganisms composed of bacteria that adhere to surfaces forming a protective matrix of extracellular substances, often diminish the effectiveness of antibiotics, increase their resistance, and contribute to the development of endodontic and periodontal diseases, ultimately causing failures of their therapies [12,13]. In teeth that are treated with root canal therapy, 77% to 90% of recurring infections and subsequent treatment failures are linked to *E. faecalis* [14–16]. Furthermore, *E. faecalis* is commonly recovered in teeth that were treated in multiple visits [12,17], likely due to its ability to form biofilms, persist in saliva, survive in nutrient-free environments, resist many antibiotics, and remain dormant as a facultative anaerobe [18–20]. In addition to root canal infections, enterococcal bacteria are also associated with endocarditis, bacteremia, urinary tract infections, intra-abdominal infections, and prostatitis [21,22]. Increased natural defense mechanisms acquired by the microorganisms, along with overprescription and increased prophylactic use of antibiotics, have elevated global concern for the emergence of antibiotic-resistant microorganisms and increased incidences of antibiotics-related secondary health issues such as dysbiosis, clostridium difficile infection, resistant urinary tract infection, and methicillin-resistant Staphylococcus aureus infections [6,23–26].

Essential oils (EOs) and their anti-inflammatory, antifungal, antimicrobial, and antibacterial properties are well elucidated [27–31]. Overall, EOs display increased sensitivity to Gram-positive bacteria, and their antimicrobial responses are mediated in part by ATP and inhibition of ATPases, disruption of membrane permeability, and inhibition of biofilm synthesis in the microbes [32–35]. In vitro studies report significant antibacterial effects of thyme, clove, sage, peppermint, lavender, cinnamon EOs, and their chemical components of thymol, eugenol, thujone, menthol, linalool, and cinnamaldehyde on caries-causing bacteria such as *streptococci* and *lactobacilli* spp. [36,37]. Among the phenylpropanoid EOs, thymol, eugenol, menthol, and cinnamaldehyde report potent antimicrobial properties in MIC (minimum inhibitory concentration), MBC (minimum bactericidal concentration), disk diffusion, and mouth rinse tests [38–41]. Moreover, combination of cinnamon, lavender, peppermint, oregano, and thyme EOs with antibiotics (β-lactam, penicillin, cephalosporin, and aminoglycoside) report enhanced and synergistic antimicrobial effects compared to when tested individually [42–46]. The enhanced antimicrobial benefits of combining antibiotics with EOs could further be examined as a novel strategy to lower the concentrations and use of antibiotics to mitigate the proliferation of antibiotic- resistant bacteria.

In this study, we assessed the presence of enhanced antimicrobial effect of cinnamon and clove essential oils when combined with penicillin (ampicillin) or aminoglycoside (kanamycin, streptomycin, gentamicin) classes of antibiotics in inhibiting the growth of *Enterococcus faecalis* using the Kirby–Bauer disk diffusion test.

## 2. Materials and Methods

### 2.1. Essential Oils and Antibiotics

Cinnamon (Cinnamomum Zeylanicum; bark) and clove (Eugenia Caryophyllus) essential oils (EOs) were purchased from Now Pure Essential Oils (Bloomingdale, IL, USA). According to the GCNS data by the manufacturer, the main chemical component of the cinnamon and clove EOs was trans-cinnamaldehyde and eugenol, respectively. The EOs were diluted with DMSO (20% *v/v*) to make a stock concentration of 10% for the cinnamon (CN) and 50% for the clove (CL) EOs. For the experiments, the CN oil was further diluted with DMSO and tested at 10%, 5%, 2.5%, and 1.25%, and the CL oil was diluted and tested at 50% and 25% concentrations. These EO concentrations were chosen as they produced a zone of inhibition that was below the CLSI Intermediate Breakpoint for *Enterococcus* spp. and when combined with an antibiotic would produce a zone of inhibition at or near the level of Intermediate Breakpoint [47].

Two classes of antibiotics, a penicillin class (ampicillin) and aminoglycoside class (kanamycin, streptomycin, gentamicin), were tested in this study. The stock concentrations of the antibiotics were: Kanamycin 50 mg/mL, Streptomycin 50 mg/mL, Gentamicin

30 mg/mL, and Ampicillin 100 mg/mL (Fisher Scientific, Waltham, MA, USA). For the experiments, the antibiotics were diluted with distilled water and tested at 10% (kanamycin 5 mg/mL, streptomycin 5 mg/mL, gentamicin 3 mg/mL, ampicillin 10 mg/mL) and at 5% (kanamycin 2.5 mg/mL, streptomycin 2.5 mg/mL, gentamicin 1.5 mg/mL, ampicillin 5 mg/mL) concentrations. These antibiotic concentrations were chosen as they produced a zone of inhibition that was below the CLSI Intermediate Breakpoint for *Enterococcus* spp., and when combined with an EO would produce a zone of inhibition at or near the level of Intermediate Breakpoint.

## 2.2. Bacterial Strain and Culture Conditions

For the study, the reference strain of *E. faecalis* (ATCC 29212) was grown and cultured on Bile Esculin Azide (BEA) agar plates. The reference strain ATCC 29212 and the BEA agar plates were purchased from Fisher Scientific (Waltham, MA, USA). Using an inoculation loop, the BEA plates were streaked with *E. faecalis* and incubated for 24 h under aerobic conditions (5% $CO_2$, 37 °C) to achieve an even growth. These cultures were used to inoculate fresh sets of BEA plates that were used for the antimicrobial susceptibility Kirby–Bauer disk diffusion test.

## 2.3. Disk Diffusion Test for EOs and Antibiotics

The Kirby–Bauer disk diffusion test was used to determine the antimicrobial susceptibility for EOs, antibiotics, and antibiotics combined with EOs on *E. faecalis*. The baseline antimicrobial effects of EOs were tested by placing 2 mL of freshly prepared CN (CN10%, CN5%, CN2.5%, and CN1.25%) and CL (CL50%, CL25%) EO solutions in individual culture tubes and vortexed for 30 s. Then, one sterile filter disk (6 mm diameter) was dropped in each tube, vortexed for another 15 s, and placed on BEA agar plates streaked with cultured *E. faecalis*. Two to three EO-soaked filter disks were placed firmly on the agar surface per each plate (n = 6–8/condition). To determine the baseline antimicrobial effects of the antibiotics, 2 mL of the 10% (kanamycin 5 mg/mL, streptomycin 5 mg/mL, gentamicin 3 mg/mL, ampicillin 10 mg/mL) and 5% (kanamycin 2.5 mg/mL, streptomycin 2.5 mg/mL, gentamicin 1.5 mg/mL, ampicillin 5 mg/mL) solutions of kanamycin, streptomycin, gentamicin, and ampicillin were added in individual culture tubes. One sterile filter disk (6 mm diameter) was dropped in each antibiotic solution, vortexed for 15 s, and placed on BEA agar plates streaked with cultured *E. faecalis*. Two to three antibiotic-soaked filter disks were placed firmly on the agar surface per each plate (n = 4/condition).

For the combinational antimicrobial effects of antibiotics and EOs, the solutions were prepared as follows: a 10% concentration of an antibiotic solution was combined with an equal volume of each of the four concentrations of EOs (CN10%, CN5%, CN2.5%, and CL100%). These pairings would yield a 5% concentration antibiotic solution containing a half dilution of EO solution that individually would produce a zone of inhibition that was below the CLSI Intermediate Breakpoint for *Enterococcus* spp., but when combined would produce ZOI at or near the CLSI Intermediate Breakpoint. For ampicillin, a 1.5 mL of 10 mg/mL ampicillin (10% *v/v*) was placed into four culture tubes that contained one of the following EO solutions: 1.5 mL of CN10%, 1.5 mL of CN5%, 1.5 mL of CN2.5% or 1.5 mL of CL100%. This 1:1 ratio of combination yielded a combined solution with final concentration of 5 mg/mL ampicillin + CN5%, 5 mg/mL ampicillin + CN2.5%, 5 mg/mL ampicillin + CN1.25%, and 5 mg/mL ampicillin + CL50%. The above steps were repeated for kanamycin 5 mg/mL (10%), streptomycin 5 mg/mL (10%), and gentamicin 3 mg/mL (10%), where 1.5 mL of antibiotics was combined with 1.5 mL of CN10%, 1.5 mL of CN5%, 1.5 mL of CN2.5%, or 1.5 mL of CL100% (Table 1). The culture tubes were vortexed for 30 s, followed by placing a sterile filter disk in each culture tube. The culture tubes were vortexed for an additional 15 s, and the filter disks were placed on BEA agar plates streaked with cultured *E. faecalis*. Two to three antibiotic-soaked filter disks were placed firmly on the agar surface per each plate (n = 8/condition).

**Table 1.** Treatment groups by combining antibiotics and EOs.

| Treatment Groups | Ampicillin 10 mg/mL | Kanamycin 5 mg/mL | Gentamycin 3 mg/mL | Streptomycin 5 mg/mL |
|---|---|---|---|---|
| **Cinnamon 10% EO** | Ampi5 + CN5% | Kana2.5 + CN5% | Genta1.5 + CN5% | Strep2.5 + CN5% |
| **Cinnamon 5% EO** | Ampi5 + CN2.5% | Kana2.5 + CN2.5% | Genta1.5 + CN2.5% | Strep2.5 + CN2.5% |
| **Cinnamon 2.5% EO** | Ampi5 + CN1.25% | Kana2.5 + CN1.25% | Genta1.5 + CN1.25% | Strep2.5 + CN1.25% |
| **Clove 100% EO** | Ampi5 + CL50% | Kana2.5 + CL50% | Genta1.5 + CL50% | Strep2.5 + CL50% |

All BEA agar plates with filter disks were secured with lab tape, inverted, and incubated at 37 °C for 24 h. The size of zone of inhibition was measured from the smallest clearings using a ruler at 1 mm scale (Figure 1). The filter disks soaked in DMSO for 15 s. were used as the control.

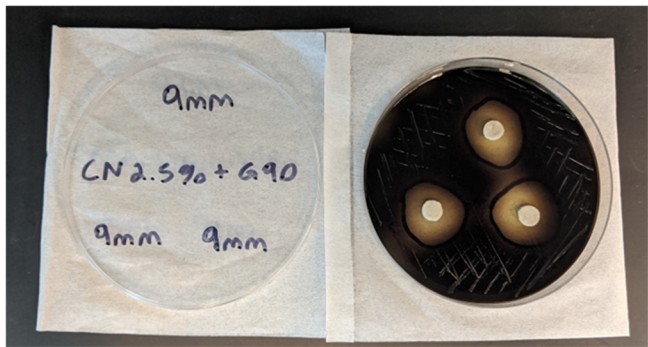

**Figure 1.** An image of BEA agar plate inoculated with *E. faecalis* showing the zones of inhibition of 5% gentamicin (1.5 mg/mL) combined with cinnamon 2.5% EO. The plate was incubated at 37 °C for 24 h and the size of zone of inhibition was measured using a ruler. BEA, Bile Esculin Azide; EO, essential oils.

*2.4. Statistical Analysis*

To determine the statistical significance in antimicrobial effects when antibiotics were combined with EOs or tested separately, the ZOIs of treatment groups were analyzed by one-way ANOVA (by treatment condition) followed by a Tukey's HSD post hoc test (GraphPad, La Jolla, CA, USA). All results were considered statistically significant at $p < 0.05$.

**3. Results**

*3.1. Antimicrobial Effects of Cinnamon and Clove EOs*

The antimicrobial efficacies of the cinnamon (CN10%, CN5%, CN2.5%, and CN1.25%) and clove (CL50% and CL25%) EO solutions were quantitated using the Kirby–Bauer disk diffusion test. The CN and CL EOs showed a concentration-dependent growth inhibition of *E. faecalis* at all tested concentrations compared to the control ($F_{(6, 24)} = 227.8$, $p < 0.0001$, Figure 2A). At the lowest concentration of CN1.25%, the zone of inhibition (ZOI) was 4.67 ± 0.21 mm. For the CN2.5% and CN5%, the ZOIs were 7.50 ± 0.22 mm and 9.00 ± 0.40 mm, respectively. At the highest concentration of CN10%, the ZOI was 11.67 ± 0.33 mm. Across all concentrations of CN EO examined, there was a significant increase in ZOI by 36% ± 1.2% as the concentration of CN EO doubled. For CL EO, the ZOIs for CL25% and CL50% were 6.00 ± 0.00 mm and 8.33 ± 0.33 mm, respectively. Similar to CN EO, the CL EO also showed a significant increase of about 38% in ZOI as the concentration increased by two-fold from 25% to 50%. When comparing the antimicrobial effects between the CN and CL EOs, the CN5% solution produced much stronger antimicrobial effect than those observed in CL 50% solution at about a 10-fold lower concentration. The diffusion disks immersed in control DMSO solution did not produce any inhibitory growth responses on *E. faecalis*.

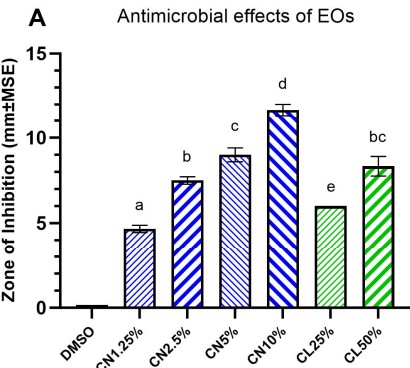 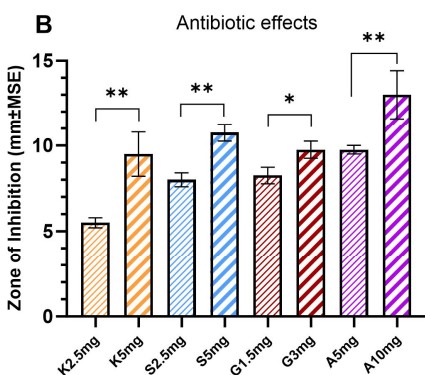

**Figure 2.** Antimicrobial effects of EOs ((**A**), cinnamon and clove) and antibiotics (**B**) on *E. faecalis* were quantitated using the disk diffusion test. The cinnamon (1.25% to 5%) and clove (25% to 50%) EOs showed significantly graded responses in inhibiting the growth of *E. faecalis* (Panel (**A**)). The cinnamon EO showed stronger antimicrobial effect than clove EO at much lower concentrations. Each antibiotic was diluted to 5% and 10% concentrations that produced a ZOI similar to that of EOs (**B**). This paired antibiotic concentration was selected for the Antibiotic + EO combination experiment. All antibiotics showed significant increase in ZOI between the 5% and 10% concentrations. CN = cinnamon, CL = clove, EO = essential oil, K = kanamycin, S = streptomycin, G = gentamicin, and A = ampicillin. Different letters (a to e) above the columns indicate significant difference between the groups ($p < 0.005$). * $p < 0.05$ and ** $p < 0.005$.

The Tukey post-hoc test showed significant paired differences between the cinnamon EOs as the concentration increased by two-fold (Figure 2A; $p < 0.005$). A similar statistical response was observed between the clove EOs where the CL50% had significantly larger ZOI than the CL25% (Figure 2A; $p < 0.001$).

### 3.2. Antibacterial Effects of Antibiotics

The effectiveness of four antibiotics (kanamycin, streptomycin, gentamicin, and ampicillin) in inhibiting the growth of *E. faecalis* was evaluated using the disk diffusion test. For each antibiotic, the stock solution was diluted in distilled water to 10% and 5% since these concentrations produced ZOI values that were similar in range with the ZOIs observed for CN and CL EOs tested previously (between 4 mm and 12 mm). The ZOI for 5% antibiotic solution of kanamycin (2.5 mg/mL), streptomycin (2.5 mg/mL), gentamicin (1.5 mg/mL), and ampicillin (5 mg/mL) were $5.50 \pm 0.28$ mm, $8.00 \pm 0.40$ mm, $8.25 \pm 0.47$ mm, and $9.75 \pm 0.25$ mm, respectively. The ZOI for 10% antibiotic solution of kanamycin (5 mg/mL), streptomycin (5 mg/mL), gentamicin (3 mg/mL), and ampicillin (10 mg/mL) were $9.50 \pm 0.64$ mm, $10.75 \pm 0.25$ mm, $9.75 \pm 0.25$ mm, and $13.00 \pm 0.70$ mm, respectively (Figure 2B). All antibiotics showed significant increase in ZOI between 5% and 10% concentration where kanamycin, streptomycin, and ampicillin showed robust increases (34% to 72%). There was significant difference among the antibiotic treatment groups ($F (7, 24) = 24.16$, $p < 0.0001$), and the Tukey post-hoc test showed significant paired differences between 5% and 10% for each antibiotic solution ($p < 0.01$).

### 3.3. Selection of Antibiotics and EOs Solution for the Combination Study on Inhibiting E. faecalis

All concentrations of cinnamon (CN10%, CN5%, CN2.5%, and CN1.25%) and clove (CL50% and CL25%) showed ZOIs that were comparable to the ZOIs of antibiotics (10% and 5%), and only CN10% solution had ZOI that was near the CLSI Intermediate Breakpoint for *Enterococcus* spp. Therefore, for the antimicrobial susceptibility test of combining EOs with antibiotics, we chose to combine the CN 5%, 2.5%, 1.25%, and CL50% for EO solutions with 5% antibiotic solutions as individually they produced the ZOIs that were below the CLSI Intermediate Breakpoint values.

### 3.4. Combined Antibacterial Effects of Antibiotics with EOs

There were significant increases in antimicrobial effects when 5% antibiotics (kanamycin (2.5 mg/mL), streptomycin (2.5 mg/mL), gentamicin (1.5 mg/mL), and ampicillin (5 mg/mL)) were combined with EOs (CN 5%, 2.5%, 1.25%, and CL50%). These enhanced antimicrobial effects were primarily observed when 5% antibiotics were combined with CN5%, where the combined solution produced ZOI that was larger than that of the corresponding 10% antibiotic solutions as well as their individual component solutions of 5% antibiotics and CN5% (Figure 3). For kanamycin, the ZOI of K2.5 mg + CN5% ($13.00 \pm 0.07$ mm; $F_{(7, 36)} = 19.44$, $p < 0.0001$) was significantly larger than the ZOI of K5 mg/mL (10% antibiotic; $9.50 \pm 0.64$ mm; $p < 0.05$) and CN5% ($7.50 \pm 0.22$ mm) and K2.5 mg/mL (5% antibiotic; $5.50 \pm 0.28$ mm) individually ($p < 0.05$). For streptomycin, the ZOI of S2.5 mg + CN5% ($12.38 \pm 0.56$ mm; $F_{(7, 36)} = 47.67$, $p < 0.0001$) was significantly larger than the ZOI of S5 mg/mL (10% antibiotic; $10.75 \pm 0.25$ mm; $p < 0.05$), and CN5% ($7.50 \pm 0.22$ mm) and S2.5 mg/mL (5% antibiotic; $8.00 \pm 0.40$ mm) individually ($p < 0.05$). For gentamicin, the ZOI of G1.5 mg + CN5% ($11.75 \pm 0.49$ mm; $F_{(7, 36)} = 44.14$, $p < 0.0001$) was significantly larger than the ZOI of G3 mg/mL (10% antibiotic; $9.75 \pm 0.25$ mm; $p < 0.05$), and CN5% ($7.50 \pm 0.22$ mm) and G1.5 mg/mL (5% antibiotic; $8.25 \pm 0.47$ mm) individually ($p < 0.05$). For ampicillin, however, the ZOI of A5 mg + CN5% ($13.00 \pm 0.39$ mm; $F_{(7, 35)} = 63.19$, $p < 0.0001$) was not significantly different than the ZOI of A10 mg/mL (10% antibiotic; $13.00 \pm 0.70$ mm), but was significantly larger than its component solutions CN5% ($7.50 \pm 0.22$ mm) and A5 mg/mL (5% antibiotic; $9.75 \pm 0.25$ mm) individually ($p < 0.05$).

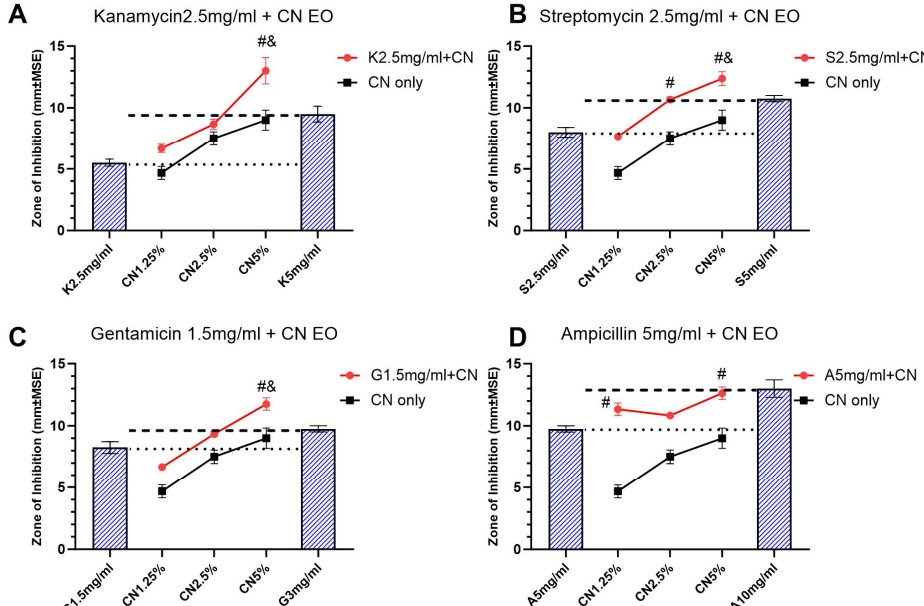

**Figure 3.** Enhanced and additive antimicrobial effects of antibiotics combined with cinnamon EO (solid red circle lines) in inhibiting the growth of *E. faecalis* in disk diffusion test. The dotted horizontal lines and shaded blue bars represent the average ZOI for 5% and 10% concentrations of antibiotics, and the solid squared lines represent ZOI for different concentrations of cinnamon EO alone. The kanamycin 2.5 mg/mL (5%) with CN5% (**A**) had significantly larger ZOI than its individual components (kanamycin 2.5 mg/mL or CN5% separately; dotted line and right solid square box) and kanamycin 5 mg/mL (10%). The streptomycin 2.5 mg/mL (5%) with CN5% (**B**) solution and gentamycin 1.5 mg/mL (5%) with CN5% (**C**) solution also produced significantly larger ZOI than their corresponding 10% antibiotics, CN5%, and 5% antibiotics separately. The ampicillin 5 mg/mL (5%) with CN5% (**D**) solution showed significantly larger ZOI only to its individual components but not to ampicillin 10 mg/mL (10%). #: $p < 0.05$ vs. 5% antibiotic concentration and corresponding CN EO individually; &: $p < 0.05$ vs. 10% antibiotic concentration alone.

For 5% antibiotic solutions combined with CN2.5%, all antibiotics failed to produce ZOI that was larger than their corresponding 10% antibiotic solutions on *E. faecalis*. Only streptomycin showed that the S2.5 mg + CN2.5% (10.67 ± 0.21 mm) had significantly increased ZOI compared to its individual component solutions of S2.5 mg (8.00 ± 0.40 mm) and CN2.5% (7.50 ± 0.22), but not to the 10% antibiotic solution. For 5% antibiotic solutions combined with CN1.25%, all antibiotics failed to produce ZOI that was larger than one of their corresponding component solutions.

For 5% antibiotic solutions combined with CL50% EO, there were no enhanced antimicrobial effects as the combined solutions did not produce ZOI that was significantly different from its individual component solutions nor from the 10% antibiotic solutions against *E. faecalis* (Figure 4).

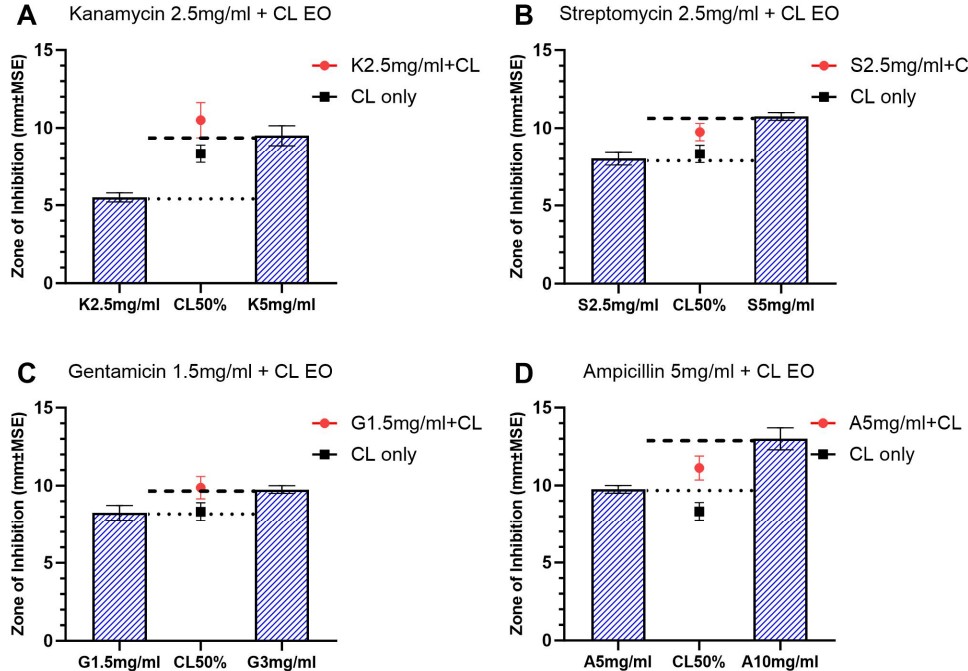

**Figure 4.** Antimicrobial effects of kanamycin (**A**), streptomycin (**B**), gentamicin (**C**) and ampicillin (**D**) combined with clove EO (solid red circles) in inhibiting the growth of *E. faecalis* in disk diffusion test. The dotted horizontal lines and shaded blue bars represent the average ZOI for 5% and 10% concentrations of antibiotics, and the solid squares represent ZOI for clove50% EO alone. Combining 5% concentration of antibiotics with CL50% did not produce any significant increase in ZOI compared to its individual components nor to 10% concentration of antibiotics against *E. faecalis*.

## 4. Discussion

The increased rate of prescription for antibiotics in recent decades, their prophylactic use, and the subsequent rise of antibiotic-resistant pathogens are changing prescription protocols for the use of antibiotics in medical and dental clinical settings and the search for alternate therapeutic medicinal compounds such as essential oils. In dentistry, endodontic diseases of periapical and intraradicular infections and their root canal treatments are becoming increasingly difficult to manage due to *E. faecalis*, a facultative aerobe that forms biofilms, survives in low-nutrient environments, and can resist antibiotics, in the root canal space. EOs, with their anti-inflammatory, antifungal, antimicrobial, and antibacterial properties, have been shown to enhance the antimicrobial effects against *E. faecalis* when combined with antibiotics and antiseptics. Given this, the examination of synergistic effects of antibiotics with EOs becomes highly relevant. In this study, we investigated the growth-inhibiting effects of penicillin and the aminoglycoside class of antibiotics when combined with cinnamon and clove essential oils in *E. faecalis*.

All concentrations of cinnamon and clove EOs were effective in inhibiting the growth of *E. faecalis* in a dose-dependent manner. The cinnamon EO produced greater antimicrobial effects than clove EO at about 10-fold lower concentration. For clove EO, the ZOI of CL25% was $7.5 \pm 0.2$ mm and increased by 139% when the concentration of clove EO was increased to CL50%. For cinnamon EO, increasing the oil concentration two-fold from 1.25% to 10% also increased the size of ZOI by about $36 \pm 1.2$% for each doubling of CN EO concentration. Our data support previous reports showing 1% to 10% of cinnamon and 50% of clove EOs were effective in inhibiting the growth of *E. faecalis* to almost 100% within 15 min of exposure, and with the cinnamon EO, the inhibitory effect was maintained for up to 10 days [48–50]. Marcoux et.al (2020) [46] reported that cinnamon EO (MIC 1.56 µg/mL) was equally effective on *E. faecalis* embedded in biofilm, killing over 90% within 15 min and outperforming chlorhexidine wash, which only showed 31% effectiveness.

For the antibiotics examined in the present study, all antibiotics were equally effective in inhibiting the growth of *E. faecalis* at 10% concentration with ampicillin having the largest ZOI. When the concentration was lowered to 5%, the streptomycin, gentamicin, and ampicillin still maintained strong antimicrobial effects against *E. faecalis*. Kanamycin 5% showed the least effectiveness with ZOI of $5.50 \pm 0.28$ mm. Our data are in line with a previous report where the MICs for ampicillin, gentamicin, and streptomycin were in a similar range of 8–14 µg/mL (MIC for kanamycin was 32 µg/mL) on an antimicrobial test against *E. faecalis*, and gentamicin was much more effective than kanamycin on an in vivo *E. coli* meningitis bacteremia test [51–53].

There were enhanced and additive synergistic antimicrobial effects against *E. faecalis* where 5% cinnamon EO combined with 5% antibiotics produced significantly larger ZOI than 10% antibiotics alone, as well as its individual components of 5% antibiotics and CN5% separately. Such additive synergistic effects were observed only with the amino-glycoside class of antibiotics tested (kanamycin, streptomycin, and gentamicin). There was also an enhanced antimicrobial effect when CN2.5% cinnamon EO was combined with 5% streptomycin (2.5 mg/mL), where the combined solution produced significantly increased antimicrobial effect comparable to that of 10% streptomycin (5 mg/mL) solution alone. Such enhancements were not observed with kanamycin/gentamicin/ampicillin combined with CN2.5% EO solution. The CN1.25% combined with 5% antibiotic solutions did not produce enhanced antimicrobial effects as their ZOI values were not significantly different than those for either one or both or their individual component solutions. For CL50% EO combined with 5% antibiotic solutions, there was no improvement in the antimicrobial effects of the combined solution beyond the effects observed for their individual component solutions.

The enhanced antimicrobial effects of combining antibiotics with cinnamon or clove EOs have been reported previously in Escherichia coli, pseudomonas aeruginosa, and in methicillin-resistant Staphylococcus aureus [54,55]. In our study, using the Kirby–Bauer disk diffusion test, we report that 5% concentration of aminoglycoside-class antibiotics (kanamycin, streptomycin, gentamicin) combined with 5% cinnamon EO produced significantly enhanced antimicrobial effect than 10% concentration of corresponding antibiotics alone against *E. faecalis*. The limitation of the present study is that since the MIC and the MBC values were not measured, it is not possible to define whether our enhanced antimicrobial observations show strictly synergistic or additive effects. However, based on our data, it may be reasonable to infer the presence of additive-like synergism where the 5% antibiotic + CN5% EO shows a significant increase in antimicrobial response compared to the 10% concentration antibiotic alone.

Acquisition of antibiotic resistance by *E. faecalis* is reported to be associated in part with its ability to synthesize β-lactamase, incorporate aminoglycoside-resistant genes aac(6′)-Ie-aph(2″)-Ia and aph(2′)-Ib, and upregulate expression of low-affinity penicillin-binding protein Pbp5, and with the presence of ATP-binding cassette multidrug efflux pump EfrAB [56–60]. These adaptations, along with its ability to survive in biologically inhospitable environments, make *E. faecalis* an ideal candidate to thrive and persist at sites in

and around endodontic infections. The antimicrobial and antibacterial properties observed in EOs, such as cinnamon and clove oils, involve disruption of bacterial genes, non-specific permeabilization of the cell membrane, and inhibition of transmembrane proton motif force and ATPase via anti-quorum sensing effects [32]. Our in vitro data show that the enhanced antimicrobial effects observed against *E. faecalis* by combining the antibiotics with cinnamon and clove EOs, presumably by interfering with the antibiotic-resistant cellular mechanisms, may be a suitable and practical approach to reduce the prevalence and incidence of persistent dental infections and treatment failures. Exploration of such strategies in in vivo and clinical studies to assess the efficacy, safety, and duration of their effects should be examined in future studies.

## 5. Conclusions

In conclusion, our study highlights the enhanced effectiveness of combining essential oils (EOs) with antibiotics in inhibiting the growth of *E. faecalis*, a pathogen associated with persistent dental infections and treatment failures. Our results show that both cinnamon and clove EOs, when tested alone, exhibited dose-dependent growth-inhibiting effects on *E. faecalis*, with cinnamon EO displaying superior efficacy at lower concentrations. Moreover, enhanced antimicrobial effects were observed when 5% cinnamon EO was combined with 5% aminoglycosides (kanamycin 2.5 mg/mL, streptomycin 2.5 mg/mL, gentamicin 1.5 mg/mL), where their combined effects were significantly stronger than the antimicrobial effects of corresponding 10% antibiotics alone (kanamycin 5 mg/mL, streptomycin 5 mg/mL, gentamicin 3 mg/mL) against *E. faecalis*. Our results demonstrate that the enhanced antimicrobial effects achieved by combining essential oils with antibiotics may be an effective strategy to maintain high antimicrobial effects while using a lower concentration of antibiotics. The antimicrobial properties of EOs by disruption of bacterial genes and cell membrane permeabilization may offer novel strategies to combat antibiotic-resistant pathogens such as *E. faecalis*. Future research should explore these strategies in in vivo and clinical settings to assess their safety, efficacy, and duration of action.

**Author Contributions:** Conceptualization, P.L., T.D. and T.Y.; methodology, P.L. and T.Y.; software (GraphPad Prism 10.0.3), J.W.L.; validation, P.L., T.D. and T.Y.; formal analysis, J.W.L.; investigation, S.J.; resources, T.D. and T.Y.; data curation, T.Y.; writing—original draft preparation, J.W.L.; writing—review and editing, J.W.L. and T.N.; visualization, J.W.L.; supervision, T.Y.; project administration, T.D. and T.Y.; funding acquisition, T.Y. All authors have read and agreed to the published version of the manuscript.

**Funding:** This research was funded by LECOM Seed Grant 23-121.

**Institutional Review Board Statement:** Not applicable.

**Informed Consent Statement:** Not applicable.

**Data Availability Statement:** Not applicable.

**Conflicts of Interest:** The authors declare no conflict of interest.

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
