# Peer review of "Potential Synergistic Inhibition of Enterococcus faecalis by Essential Oils and Antibiotics"

_applsci, doi:10.3390/app131911089_

Round 1
Reviewer 1 Report
The authors should clearly explain how the synergistic effect was determined. It seems that the results could only be of a synergistic type, and not of another type such as antagonism or additivity.
Explain in detail the methodology and the statistical methods used to determine these effects.
Why isobolograms and interaction index were not used to evaluate these effects?
After explaining this, authors should include literature that supports these types of interactions or effects according to the above.
Author Response
September 29, 2023
To: MDPI Applied Sciences
Ms. Aryal Song, Assistant editor
RE: Manuscript ID: applsci-2628876
Title: Synergistic Inhibition of Enterococcus faecalis by Essential Oils and Antibiotics: Implications for Enhanced Root Canal Therapy
Dear Ms. Aryal Song,
We have received comments for our manuscript from three reviewers on Sep 20th and 25th 2023. We have made some changes in the text of our manuscript based on their comments. We thank the reviewers for their valuable input, and are attaching our responses addressing specific comments from each reviewer:
Comments by Reviewer 1:
- The authors should clearly explain how the synergistic effect was determined. It seems that the results could only be of a synergistic type, and not of another type such as antagonism or additivity.
The reviewer is correct that we are unable to strictly define synergism in our study as the values for MIC and MBC were not examined in this study. However, our revised data (Fig. 2) would show that combining a lower dose of antibiotic with cinnamon essential oil produced antimicrobial effect that was greater than having used a two-fold increased concentration of antibiotics alone. Therefore, we have revised our wording from “synergistic effect” to “enhanced antimicrobial effect” reflected both in the title of the manuscript as well as in the discussion section.
- Explain in detail the methodology and the statistical methods used to determine these effects.
The Materials and Methods section was revised to include descriptions for preparing antibiotic/essential oil solutions and steps taken to make the combined antibiotic+essential solutions.
All statistical analysis were performed using One-way ANOVA (GraphPad Prism v.10) by treatment groups as described, so no additional text was added in the Statistical section of the manuscript.
- Why isobolograms and interaction index were not used to evaluate these effects?
Our study measured the zone of inhibition by disk diffusion test but did not quantitate the MIC and MIC values to calculate the FICI using the broth dilution or checkerboard assay. As such we are unable to generate an isobologram. However, we have revised our Fig. 3 to show enhanced antimicrobial effect of combined 5% antibitoc+EO solution compared to using a higher (10%) concentration of antibiotic solution alone.
- After explaining this, authors should include literature that supports these types of interactions or effects according to the above.
We have included references #54 and 55 to indicate values for zone of inhibition (from disk diffusion test) are used to show increased antimicrobial effects of combining cinnamon essential oils with antibiotics on Escherichia coli, pseudomonas aeruginosa and in methicillin resistant Staphylococcus aureus.
Reviewer 2 Report
The presented study is interesting and offers a possible solution in the treatment of infections caused by Enterococcus faecalis. Even so, I have the following comments that need to be addressed:
Biofilm is mentioned several times in the study, but your research is based exclusively on a reference strain, therefore it is necessary to supplement the research with a strain directly from clinical samples of root canal infection. If this is not possible, then the name of the manuscript should be changed, because this study could be implemented in other branches of medicine, not only in dentistry. Therefore, the title of the post needs to be considered and redesigned. In addition, Enterococcus faecalis is written in italics in the name as well. It is also necessary to check in the text.
Introduction- long, I will focus more on enterococci than on other types of microorganisms
Line 47- you mention a biofilm with bacteria and fungi...if there are mixed biofilms, it is a problem to claim that the given materials will form a biofilm...develop or modify the sentence so that the mentioned fungi are not there
Methodology - Methodology - why did you choose these concentrations of antibiotics?– explain in one or two sentences
- Why did you choose clove and cinnamon and these concentrations?....it is necessary to explain
- Why did you use the disk diffusion method and not the microdilution method, which would have made it more precise what concentrations would be appropriate?
265 – "previous study"... must be cited if it is published, if not, rewrite the sentence
Discussion - write less about biofilm testing in the discussion, since in your study the effect was only tested on a pure reference strain, where the antibacterial effectiveness may differ from biofilm. In addition, biofilm from clinical samples is more often multispecies.
Author Response
September 29, 2023
To: MDPI Applied Sciences
Ms. Aryal Song, Assistant editor
RE: Manuscript ID: applsci-2628876
Title: Synergistic Inhibition of Enterococcus faecalis by Essential Oils and Antibiotics: Implications for Enhanced Root Canal Therapy
Dear Ms. Aryal Song,
We have received comments for our manuscript from three reviewers on Sep 20th and 25th 2023. We have made some changes in the text of our manuscript based on their comments. We thank the reviewers for their valuable input, and are attaching our responses addressing specific comments from each reviewer:
Comments from Reviewer 2:
- The presented study is interesting and offers a possible solution in the treatment of infections caused by Enterococcus faecalis. Even so, I have the following comments that need to be addressed:
- Biofilm is mentioned several times in the study, but your research is based exclusively on a reference strain, therefore it is necessary to supplement the research with a strain directly from clinical samples of root canal infection. If this is not possible, then the name of the manuscript should be changed, because this study could be implemented in other branches of medicine, not only in dentistry. Therefore, the title of the post needs to be considered and redesigned. In addition, Enterococcus faecalis is written in italics in the name as well. It is also necessary to check in the text.
The present study examined a reference strain of E. faecalis (ATCC 29212). The title is revised to remove reference to “dental implications” and the manuscript text is revised to indicate a reference strain of bacteria (E. faecalis) was used in the study.
The manuscript is reviewed to italicize E. faecalis.
- Introduction- long, I will focus more on enterococci than on other types of microorganisms
Development of antibiotic resistant microorganism is universal, including to E. faecalis. The text included examples of other types of microorganisms that also produce biofilms and show resistance (or development of resistance) to antibiotics. Therefore, the text referencing other microorganism was left as is.
- Line 47- you mention a biofilm with bacteria and fungi...if there are mixed biofilms, it is a problem to claim that the given materials will form a biofilm...develop or modify the sentence so that the mentioned fungi are not there
The text “fungi” was removed from the paragraph
- Methodology - Methodology - why did you choose these concentrations of antibiotics?– explain in one or two sentences
- Why did you choose clove and cinnamon and these concentrations?....it is necessary to explain
Based on the CLSI Breakpoints for enterococcus spp, we chose to produce values that were below the Intermediate Breakpoints by EOs-alone or antibiotics-alone, and when combined would produce a value that would reach the Intermediate Breakpoints.
A description of this effect is included in the Materials and Methods section.
- Why did you use the disk diffusion method and not the microdilution method, which would have made it more precise what concentrations would be appropriate?
This was an initial proof-of-concept study. The authors are planning a follow up study where broth dilution or checkerboard assay will be employed.
- 265 – "previous study"... must be cited if it is published, if not, rewrite the sentence
The “previous report” is citing references #50-52
- Discussion - write less about biofilm testing in the discussion, since in your study the effect was only tested on a pure reference strain, where the antibacterial effectiveness may differ from biofilm. In addition, biofilm from clinical samples is more often multispecies.
The biofilm-producing microorganisms such as E. faecalis is a challenge to the field of endodontic. Although current study examined the reference strain, this study focused on the importance of developing methods to inhibit their growth and at the same time use a lower dose of antibiotics to prevent development of antibiotic resistant strains. The topic of Biofilm was the basis for this study and future studies that are planned. Therefore, we have left the references of biofilm in the discussion section as is.
Author Response
September 29, 2023
To: MDPI Applied Sciences
Ms. Aryal Song, Assistant editor
RE: Manuscript ID: applsci-2628876
Title: Synergistic Inhibition of Enterococcus faecalis by Essential Oils and Antibiotics: Implications for Enhanced Root Canal Therapy
Dear Ms. Aryal Song,
We have received comments for our manuscript from three reviewers on Sep 20th and 25th 2023. We have made some changes in the text of our manuscript based on their comments. We thank the reviewers for their valuable input, and are attaching our responses addressing specific comments from each reviewer:
Comments from Reviewer 3:
Congratulations to the authors for the effort to find alternative ways to reduce the use of antibiotics. The ideal would be the permanent replacement of antibiotics.
Remarks:
- line 63 italics of the microbial strain
Corrected.
- range of lines 72-74 - the bibliography should be presented more specifically by types of actions and not so globally. This observation can be extended for the entire article, because the presentation of the bibliography in a wide range is very difficult for the reader to follow.
Where possible, the authors tried to reference direct experimental research articles rather than review articles. And even though there are numerous articles that describe about the effectiveness of essential oils, the authors include only those from well-known and cited journals.
- line 89 - the initial concentration of the monitored substances must be specified and if it can present variations, for various reasons (species, geographical area, etc. or bibliography)
We were limited by the data provided by the manufacturer Now Pure Essential Oils. We were able to obtain their GCMS data, but the manufacture was unable to provide the analytic data for the oils. The manufacturer describes their product as being 100% pure oil.
- line 105 - the authors referred to some treatment conditions, but did not specify what they are!
The Materials and Methods section is revised to include details for the dilution and preparation of solution samples.
- range of lines 121-124 or in table 1 - it must be specified that % are volumetric, when specifying the unit of measurement of concentrations
The revised Materials and Methods include description of 1.5ml of essential oil combined with 1.5ml of antibiotic solution (at 1:1 ratio) and the resultant mixture would contain one-half dose of individual solutions. E.g. 1.5ml of 5% kanamycin was combined with 1.5ml of 10% cinnamon oil to form a 3ml of 2.5% kanamycin with 5% cinnamon oil.
- line 147 - what does DMEM represent?
This was corrected to DMSO
- line 267 - italics of the microbial strain
The E. coli is italicized
In figure 1, the holographic notations (written by hand) do not agree with the notations in the article. I recommend removing the holographic writing and introducing more examples of images with inhibition zones. Note: in the version uploaded by the authors of the image, the microbial field developed on the culture medium is not visible, in order to correctly highlight the zone of inhibition!
The writing on plate lid “G90” indicates that gentamycin was diluted “down to 90%” or 10% final concentration.
The uploaded image shows the halo of zone of inhibition in white background. All images from the study shows similar tonality, so we have not replaced Fig. 1 with another image.
In figure 2, the * and ** type correlations are missing in graphs B, C and D. Why?
We have revised Fig. 2 in new format. The revised graph also uses * above a bracket indicating a paired-wise significance between indicated treatment conditions.
The authors did not comment on why there are no # or ## type correlations for all the graphs considered. DiƩo for Figure 3.
Figure 3 is also revised to show presence of statistical significance where available.
For both figure 2 and figure 3, I recommend a narrower division for the graph's ordinate (Oy axis) in order to support the statements in the text.
Figures were revised to include smaller subdivision for the Y axis
The notations of the samples are not maintained: e.g. G1.5 and G1.5mg, and if working with other units of measure, I recommend maintaining a single way of using the unit of measure for concentrations, so that the results presented in the discussions are supported and correctly understood.
The figure legends were changes to uniformly describe as G1.5mg/ml
Round 2
Reviewer 1 Report
The authors must explain how they calculated the concentrations that were used in each combination.
Using the diameters of the inhibition zones, the minimum inhibitory concentrations can be calculated and subsequently test combinations using concentrations obtained experimentally for each drug separately. Subsequently, construct the isobolograms. You should consider carrying out more experiments.
Author Response
The authors greatly appreciate the valuable comments from the manuscript reviewers. We are including our responses to the review comments.
Comments from Reviewer 1 – round 2
The authors must explain how they calculated the concentrations that were used in each combination.
We have revised and expanded the Materials and Methods section to describe on how the combined antibody-EO solutions were made and used. In this paragraph, we have also included the rationale behind selecting these specific concentrations of antibiotics and EOs as well as the process on how these individual solutions were made. The revised sentences and paragraphs are found on lines 93-96, 104-107, and 135-153.
Using the diameters of the inhibition zones, the minimum inhibitory concentrations can be calculated and subsequently test combinations using concentrations obtained experimentally for each drug separately. Subsequently, construct the isobolograms. You should consider carrying out more experiments.
As described by the reviewer, it is possible to use the published reference data to estimate the MIC for our antibiotics and EOs against Enterococcus spp. and to construct isobolograms. This approach, however, may be an inference (indirect comparison measurement) rather than actual dose responses measured by the present study. The present study did not perform broth dilution or checkerboard assay to experimentally calculate MIC and FICI to be able to show synergism using an isobologram. Therefore, we have revised the figures 3 and 4 to illustrate enhanced antimicrobial effects of combining an antibiotic with an EO, instead.
In the design of future studies, the broth dilution and/or checkerboard assays will be included, along with disk diffusion tests, to specifically quantitate and determine the presence of synergism or additive effects.

Reviewer 2 Report
I have no comments.
Author Response
Comments from Reviewer 2 – round 2
I have no comments.
The authors greatly appreciate the valuable comments from the manuscript reviewers.